# A Simplified Murine Model to Imitate Flexor Tendon Adhesion Formation without Suture

**DOI:** 10.3390/biomimetics7030092

**Published:** 2022-07-07

**Authors:** Rong Bao, Shi Cheng, Jianyu Zhu, Feng Hai, Wenli Mi, Shen Liu

**Affiliations:** 1Department of Orthopaedics, Sixth People’s Hospital, Jiao Tong University, 600 Yishan Rd, Shanghai 200233, China; 17301050001@fudan.edu.cn; 2Department of Integrative Medicine and Neurobiology, School of Basic Medical Science, Institutes of Integrative Medicine, State Key Laboratory of Medical Neurobiology and MOE Frontiers Center for Brain Science, Institutes of Brain Science, Shanghai Medical College, Fudan University, Shanghai 200032, China; 17301050003@fudan.edu.cn (S.C.); 17111520012@fudan.edu.cn (J.Z.); 17301050055@fudan.edu.cn (F.H.); 3Shanghai Key Laboratory of Acupuncture Mechanism and Acupoint Function, Fudan University, Shanghai 200032, China

**Keywords:** flexor tendon, tendon adhesion, biomechanics, collagen deposition

## Abstract

Peritendinous adhesion (PA) around tendons are daunting challenges for hand surgeons. Tenotomy with various sutures are considered classical tendon repair models (TRM) of tendon adhesion as well as tendon healing. However, potential biomimetic therapies such as anti-adhesion barriers and artificial tendon sheaths to avoid recurrence of PA are sometimes tested in these models without considering tendon healing. Thus, our aim is to create a simplified model without sutures in this study by using three 6 mm longitudinal and parallel incisions called the longitudinal incision model (LCM) in the murine flexor tendon. We found that the adhesion score of LCM has no significant difference to that in TRM. The range of motion (ROM) reveals similar adhesion formation in both TRM and LCM groups. Moreover, mRNA expression levels of collagen I and III in LCM shows no significant difference to that in TRM. The breaking force and stiffness of LCM were significantly higher than that of TRM. Therefore, LCM can imitate flexor tendon adhesion formation without sutures compared to TRM, without significant side effects on biomechanics with an easy operation.

## 1. Introduction

Peritendinous adhesion (PA) as a complication of tendon rupture or surgery seriously limits patients’ daily activity [1,2]. It can occur in almost every kind of injured tendon [3], and its incidence was estimated to be about 221 injuries per 100,000-person years in the US. Among these, up to 30–40% of cases led to disability, resulting in financial burden worth over billions of dollars in healthcare each year in the US [4,5]. Thereafter, lots of studies focused on the mechanisms and treatment of PA [2,6]. Among them, classical murine flexor tendon adhesion models (TRM) were widely used to imitate human PA which contained tendon transection and subsequent suture repair [7,8]. However, clinically, there were lots of patients suffering from PA without complete tendon cut-off. But few animal models without sutures were designed to imitate human adhesion formation in this case. For example, only three in 26 murine flexor tendon models used models without sutures according to a recent review [2].

Tendons can be classified into both intrasynovial and extrasynovial, according to whether there are synovial sheathes on their surfaces. For example, the Achilles tendon, patellar tendons and plantar flexor digitorum longus (FDL) tendon all belong to the extrasynovial type [9,10,11], while the flexor digitorum profundus belong to the intrasynovial type [12]. Animal models relating to scientific studies in these two types of tendons, are available [9,10,11,12,13,14,15,16]. They provide the possibility to replicate PA after tendon injury [17,18]. They also represent a unique tool to investigate future interventions and fundamental mechanisms of PA that cannot be directly performed in humans [19,20,21,22] though the tendon in animal models is usually different to humans in anatomy. The model in our experiments were aimed at the FDL tendon.

Anatomically, a tendon is arranged orderly, in a hierarchical manner, along the main axis of the tendon which includes six levels: collagen molecule, pentafibril, collagen fibril, collagen fiber, fascicle and whole tendon. Thus, the transverse tenotomy obviously has side effects on tendon healing, while longitudinal incisions in tendon have few influences on tendon healing. Moreover, the tendon is covered by the epitenon which is connected to the interfascicular matrix [23]. Taylor et al. pointed out that a defect in the epitenon is required for tendon adhesion formation, which means even if tendons are not fully injured or cut-off, adhesion will still form [6]. Thereafter, the peritendinous formation may be due to the damage to the epitenon of the tendon. 

Tenolysis is one of the most common treatments for PA. Medicine therapy and cell therapy are also used in tendon adhesion [9,10,11,12,13,14,15,16]. Meanwhile, synthetic materials such as biomimetic anti-adhesion barriers and engineered tendon sheaths are used to prevent recurrence of adhesion [2]. Before being applied in the clinic, such new strategies are usually investigated in classical murine models like TRM and transgenic mice with classical models which are widely used due to the advances in in vivo mechanistic studies [24,25]. However, classical murine models contain different defect sizes of tendons and various types of sutures. In these situations, there are many discrepancies in the verification of new strategies. Additional evaluation tests such as breaking force should also be used. Furthermore, it is a highly skilled and complicated procedure to repair the transected tendon by using sutures under a microscope. Besides, a high rate of postoperative rupture is another deficiency in the classic murine PA model [19,24,26] Thus, it is crucial to design a simplified murine model to imitate human adhesion formation without tendon transection and subsequent suture.

Based on our previous studies on TRM [27,28], in this study we designed a simplified murine flexor tendon adhesion model easily created by longitudinal plantar incisions in the flexor digitorum longus (FDL) tendon. We hypothesized that LCM achieved similar adhesion compared with TRM without significant side effects on the breaking force of the tendon. To validate this hypothesis, histological analysis, mechanical analysis and real-time RT-PCR were performed, respectively.

## 2. Materials and Methods

### 2.1. Processing of Longitudinal Incision Model

Animals were obtained commercially from Shanghai SLAC Laboratory Animal Co Ltd. and studies were approved by the Ethics Committee of Shanghai Sixth People’s Hospital (Reference Number: 2021-0839). The experimental animals were divided into 3 groups by a random chart method. The 3 groups were: the LCM group, TRM group and control group. We used 8–10-week-old male C57BL/6 mice because of their homogeneity and our clinical experience that males who were manual workers or outdoor workers were more likely to suffer tendon impairment. A total of 6 mice in every group were used for H&E staining. In every group 8 mice were used for the breaking force test. In every group 8 mice were used for ROM and gliding coefficient measurement. In every group 10 mice were used for PCR. There were 1–2 mice that died by accident in some groups before euthanasia. Mice were anesthetized by ketamine (60 mg/kg body weight) and xylazine (4 mg/kg body weight). After anesthetization, the flexor digitorum longus (FDL) tendon in the right hind-paw, was surgically explored through posterior midline incision. To create LCM, briefly, the FDL tendon was cut along three 6 mm longitudinal plantar incisions. To create TRM, the FDL tendon was transected and repaired using Kessler sutures in the right hind-paw (Figure A1), [19,20,21,22,24,25,26]. To create a control group the C57BL/6 mice with sham surgery in the right hind-paw were treated, which meant they had the same processing as the other two groups except for tendon impairment. Then the vincula tendinums of 3 groups were removed during modeling. Subsequently, the proximal FDL tendons of the mice in all groups were isolated and transected at the myotendinous junction in order to reduce the influence of movement on the adhesion formation, which also ensured the homogeneity of this study. The mice were closed with a 6-0 suture allowing them to move freely in their cages after the wound. The mice were humanely euthanized by intraperitoneal injection of pentobarbital (100 mg/kg) after 14 days. 

### 2.2. Histological Analysis

The right hind-paws were harvested 14 days after surgery (*n* = 6 for each group) by disarticulating the intact feet and tibias at the ankle joints. Specimens were treated by fixation, decalcification and embedding. Briefly, intact feet were fixed in 4% neutral buffered paraformaldehyde solution (Beyotime, Shanghai, China), and then decalcified in 10% EDTA (Aladdin, Shanghai, China) at 4 °C for 28 days. Subsequently, the decalcified feet were dehydrated in 30% sugar water and then embedded in OCT solution. Subsequently, 10μm frozen slices were cut along the longitudinal axis of the FDL tendon, and stained with Hematoxylin-Eosin (H&E) Staining (Solarbio, Beijing, China). The widely-used semiquantitative scoring system, is a typical method to histologically evaluate the adhesions formed [27,29,30,31,32]. The histological scoring system for adhesions were scored in grades 0–10 (Figure A2), and the histological scoring system for healing was scored into grades 0–3 in the field of cellularity, parallel fibers, and fiber continuity (Figure A3) [32,33,34]. 

### 2.3. The Range of Motion and Gliding Coefficient

The range of motion, (*n* = 8 for LCM group; *n* = 7 for TRM group; *n* = 6 for control group) namely assessment of metatarsophalangeal (MTP) joint flexion and gliding coefficient upon FDL tendon loading, was measured following the method contrived by Sys Hasslund et al. [34]. Briefly, the injured hind-limb was harvested and fixed in a custom apparatus where the tibia was rigidly gripped to prevent rotation (Figure A4). Then two different loadings (0 g/19 g) were respectively suspended on the proximal end of FDL statically. With each loading, a digital image was taken. The difference of MTP joint angles between two images (0 g/19 g) was the range of motion. Gliding coefficient was calculated by the formula:(1)$$M=\beta \times \left(1-e^{-\frac{m}{\alpha }}\right)\;(R^{2}=0.093\pm 0.07,\;p<0.05)$$
*M*: MTP Flexion Angle *β:* maximum flexion angle *m:* weight of loading (19) *α:* gliding coefficient

### 2.4. Gene Expression Using Real-Time PCR

To determine the collagen I and III deposition in PA tissues, RNA isolation and quantitative real-time PCR were performed on the specimens from the LCM (*n* = 10), TRM group (*n* = 9) and control group (*n* = 10). In brief, the PA tissues around the injured tendon after harvest were retrieved and immediately frozen in liquid nitrogen, which were stored separately from each mouse. Total RNA was extracted with the TRIzol^®^ Reagent (Invitrogen Corporation, Carlsbad, CA, USA), which was then immediately reverse transcribed with a reverse transcription kit (RR036A, TaKaRa, Otsu, Japan) and stored at −20 degrees. SYBR Green PCR Master Mix (RR420A, TaKaRa) was used for quantitative real-time PCR (Table A1). The mean cycle threshold (Ct) values were used to calculate the gene expression standardized to GADPH expression as an internal control. 

### 2.5. Biomechanical Analysis

The FDL tendon of the right hind-paw was retrieved for biomechanical analysis (*n* = 8 for LCM group; *n* = 6 for TRM group; *n* = 7 for control group) according to previous studies [28]. Briefly, the injured tendon was identified from surrounding tissue at the myotendinous junction, freed along the axis of the tendon and cut at the ends. Subsequently, both ends of the tendon were secured for direct gripping in the mechanical test. The proximal terminal was then pulled under tension in a displacement control at a rate of 30 mm/min until failure to obtain a force-displacement curve (Universal Testing System, instron 5569). Breaking force and stiffness were recorded automatically by the system, and stiffness was the slope of the linear region of the documented curve. Failure modes were carefully observed and recorded.

### 2.6. Statistical Analysis

All statistical analyses were performed using Prism, software. Data were expressed as the mean ± standard deviation. One-way ANOVA for comparisons between three groups were used to determine the values. Significant difference was considered as *p* < 0.05.

## 3. Results

### 3.1. Histological Analysis of Longitudinal Incision Model

Tendon samples were processed as described above and harvested at 14 days (Figure 1). It is well recognized that histologic scoring grades can provide convincing evidence to investigate the extent of adhesion formation and tendon healing. Based on the pictures of H&E staining, PA tissues were around the injured tendon (Figure 2A–C) in LCM and TRM groups but not the control group. The suture site was especially detected in the TRM. Hypercellular fibrotic scar tissue can be investigated in the PA sites of tendons in LCM and TRM. No significant difference was detected between TRM and LCM in the score of tendon adhesion and no significant difference was detected between LCM and control in the score of tendon healing except for parallel fibers, which means LCM can imitate the clinical adhesion after tendon injury as TRM does, and has little influence on tendon healing (Figure 2D–G).

### 3.2. ROM of Longitudinal Incision Models

The range of motion and gliding reveal the extent of tendon adhesion. From the results, no significant difference was detected between the two groups but they both had significant differences to the control group (Figure 3).

### 3.3. Gene Expression in Longitudinal Incision Model

It is well known that collagen I and collagen III are crucial components of PA tissues. Collagen I and III are common proteins which can be tested in PA tissues. Briefly, the quantity of collagen III accumulated during the early period of tendon adhesion. Later, it is replaced by collagen I. Thus, collagen I and III can reveal tendon formation. We tested their mRNA expression to verify that LCM imitated tendon adhesion as TRM did [3,35,36,37,38]. Thus, the gene expression of collagen I and collagen III represent the process of adhesion formation. Based on the real-time PCR results, we learned that the mRNA expression levels of collagen I and III in the LCM group had no significant difference with those in the TRM group, (Figure 4).

### 3.4. Biomechanics of Longitudinal Incision Model

Biomechanical analysis can record the peak breaking forces of tendons, which reflects tendon healing. Representative force-displacement curves of the TRM group and LCM group are shown in Figure 5A–C. Breaking forces (Figure 4D) were directly investigated. From the results, it was concluded that the breaking force in the LCM group was significantly larger than that in the TRM group but had no significant difference with the control group. Moreover, the stiffness was calculated by the testing system and represents the slope of the linear region of the documented curve (Figure 5E). The stiffness in the LCM group was markedly larger than that in the TRM group but had no significant difference with the control group. 

## 4. Discussion

PA is complicated by tendon rupture and usually results in disability. Thus, animal models are widely used to investigate potential anti-adhesion therapies and tendon healing [35,36]. In this study, a novel longitudinal incision model without tendon cut-off and suturing was created by three 6 mm longitudinal and parallel incisions in the murine flexor tendon. Similar adhesion formations in both TRM and LCM groups were observed based on H&E sections as associated semiquantitative histological scoring showed no significant difference between the LCM and TRM groups. The range of motion and gliding coefficient also revealed similar adhesion formations in both the TRM and LCM groups. On the basis of Real-Time PCR, mRNA expression levels of collagen I and III in the LCM group had no significant difference to that in the TRM group. In addition, the breaking force of the LCM group was significantly larger than that of the TRM group and had no significant difference to the control group, which suggests LCM formed adhesions without tendon transection, despite similar evaluated results in the tendon healing score of parallel fibers with the TRM group. The difference between the LCM and control group in parallel fiber score can’t be avoided because the impairment of the LCM caused damage of the fiber and initial fiber regeneration is disordered. But the breaking force showed that this difference had no influence on the healing of the whole tendon. A likely explanation was that the LCM had many intact fibers beside the incisions and these intact fibers can provide enough breaking force. The LCM group had a good biomechanical result because the longitudinal incisions maintained the continuity of the tendon body. 

There are many methods in processing animal tendons to imitate peritendinous adhesion. Among them, the most widely used animal model is referred to as the classical murine flexor tendon adhesion model, created by transverse tenotomy and subsequent suture repair to imitate human PA [2]. Furthermore, Lyras et al. applied part of the transection without sutures in the Achilles tendon [37]. Fukawa et al., even applied full transection without sutures in the Achilles tendon [38]. All of these models were cut off transversely, resulting in potential transverse rupture of tendon fibers and a potential decrease in tendon strength during the period of tendon healing. The ROM, an important evaluation tool of tendon adhesion, is unable to be tested if the tendon is finally ruptured. It also causes difficulty in evaluating adhesion scores because of the inconsecutive tendon. In addition, the models in previous studies used various modified sutures after tenotomy [2]. However, the modified sutures had a risk of rupture and were required to be processed under microscopy, which was difficult to handle. In this study, a novel simplified murine flexor tendon adhesion model was easily created by longitudinal plantar incisions in the FDL tendon, which ensured a reasonable direction according to the anatomy and avoided the heterogeneity caused by sutures. Similar adhesion and healing formation was shown in the LCM and TRM, which indicated that our murine FDL tendon model provided an easier method to generate tendon adhesion without worrying about complex suture techniques with microscopy. Especially, according to the results of biomechanical analysis, the breaking force of the LCM group was significantly larger than that of the TRM group and similar to the normal tendon due to the longitudinal injury to the tendon in the LCM group. Thereafter, LCM can meet the needs of verification of new strategies without concern for the failure of tendon healing. 

We have to admit that this new model has some inherent limitations. To quantify the relationship between in-depth effects of incisions in the FDL tendon and severity of adhesion, different kinds of incisions such as various lengths, depths and numbers of incisions in the FDL tendon, still require further exploration. According to our experimental design, this model can be easily repeated and standardized, even without the help of microscopy. Moreover, we have managed to introduce a simple kind of adhesion model, which may enhance the efficiency of research in tendon adhesion. There is a huge potential for this simplified model to be suitable for studies focusing on tendon adhesion without the possibility of tendon rupture.

## 5. Conclusions

Three 6 mm longitudinal and parallel incisions in the murine flexor tendon can result in PA. Furthermore, there are no significant differences between the LCM and TRM groups in semiquantitative histological scoring, range of motion, gliding coefficients and mRNA expression levels of collagen I and III. However, the breaking force of the LCM group was significantly larger than that of the TRM group and had no significant difference to the control group. Thus, the LCM may have potential value in future biomimetic investigations which imitate clinical tendon adhesion without flexor tendon transection and subsequent sutures, due to its easy operation and reliability.

## Figures and Tables

**Figure 1 biomimetics-07-00092-f001:**
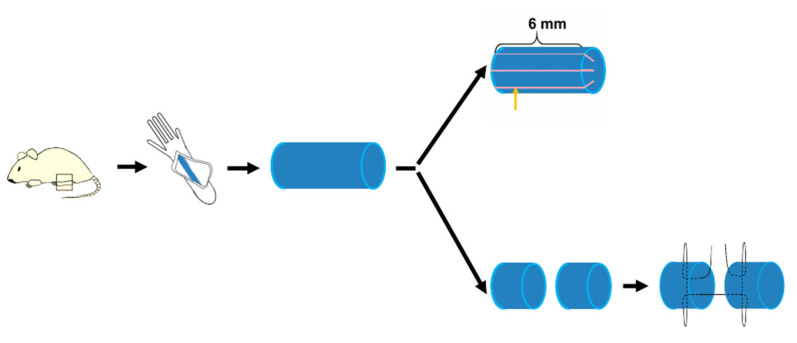
**The schematic diagram of the longitudinal incision model and classical tendon repair model.** It includes anesthesia and exposure of the FDL tendon. Three 6 mm longitudinal plantar incisions were made on the LCM. Yellow arrow shows the incision. TRM consists of tenotomy and Kessler suture.

**Figure 2 biomimetics-07-00092-f002:**
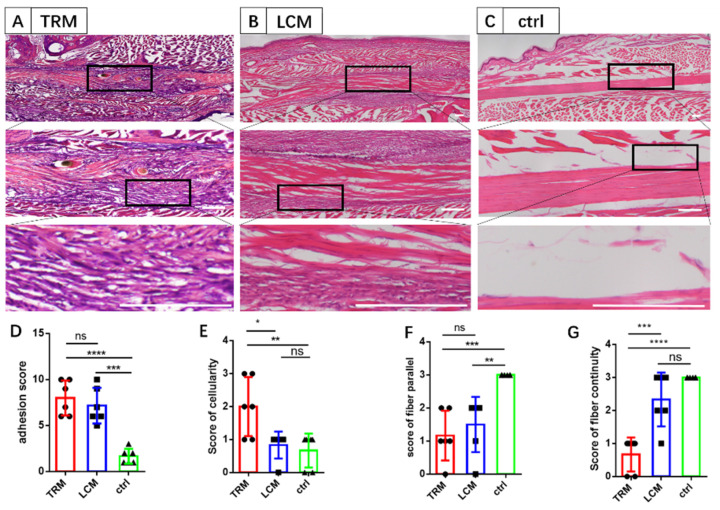
**The evaluation of tendon adhesion and healing in the three groups.** (**A**–**C**) The H&E staining showing space between the tendon and its surrounding tissue. (**D**) Tendon adhesion score. (**E**–**G**) Tendon healing scores including the cellularity score, parallel fiber score and fiber continuity score. *n* = 6 mice for all groups. Scale bars, 200 μm. Mean ± SD, * *p* < 0.05; ** *p* < 0.01; *** *p* < 0.001; **** *p* < 0.0001; ns, no significance.

**Figure 3 biomimetics-07-00092-f003:**
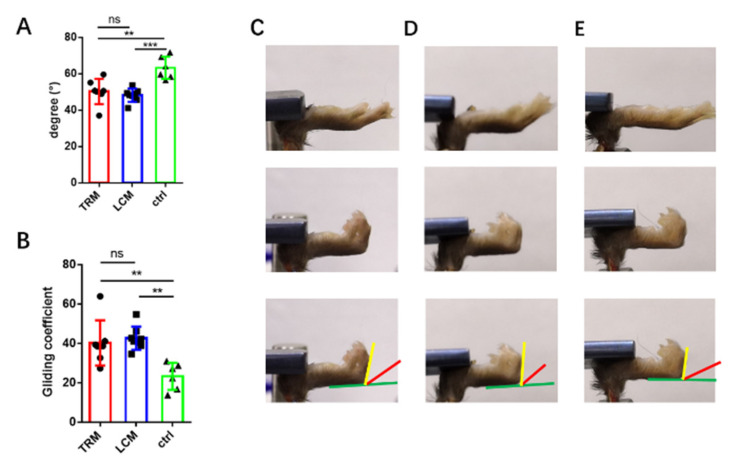
**The range of motion in the three groups.** (**A**,**B**) The range of motion (ROM) and gliding coefficient. (**C**–**E**) The Schematic diagram of measuring ROM. (**C**–**E**) The representative measurement of ROM in the LCM, TRM and control group, respectively. Two different loadings (0 g/19 g) were respectively suspended on the proximal end of FDL statically (upper subfigures were 0 g and middle subfigures were 19 g). The difference of MTP joint angles between the two images (0 g/19 g) was the range of motion (ROM) as the lower subfigures showed; green line was the reference, red line and yellow line were MTP joint angles under two different loadings, respectively. The angle made by the red and yellow line was the ROM. The examples of the ROM in (**C**–**E**) were 44.1°, 46.4° and 57.7°. Mean ± SD, ** *p* < 0.01; *** *p* < 0.001; ns, no significance.

**Figure 4 biomimetics-07-00092-f004:**
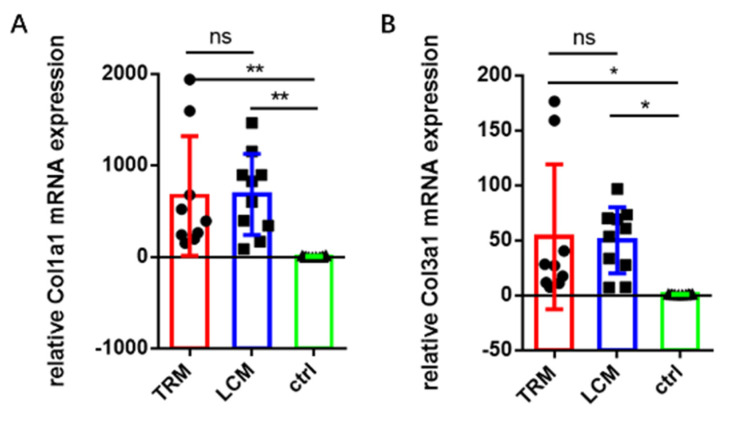
**Relative mRNA expression of collagens related to tendons in the three groups.** (**A**,**B**) Relative mRNA expression of collagen I and III. Mean ± SD, * *p* < 0.05; ** *p* < 0.01; ns, no significance.

**Figure 5 biomimetics-07-00092-f005:**
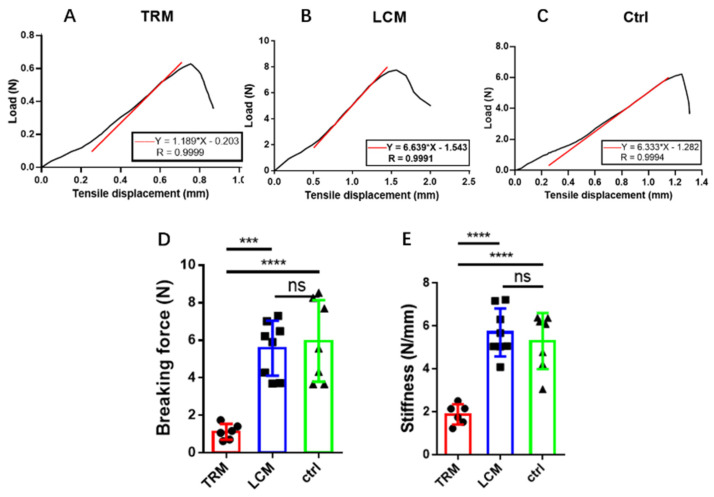
**Biomechanics of the three groups.** (**A**–**C**) Representative force-displacement curve. Breaking force and stiffness were recorded by the testing system automatically. The diagrams were copied from the testing system by the Engauge Digitizer and Prism, software. Red lines show the slope of the linear region of the documented curve. (**D**) Breaking force. (**E**) Stiffness. Mean ± SD, *** *p* < 0.001; **** *p* < 0.0001; ns, no significance.

## Data Availability

Not applicable.

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
