# Peer review of "A Simplified Murine Model to Imitate Flexor Tendon Adhesion Formation without Suture"

_biomimetics, 2022, doi:10.3390/biomimetics7030092_

Round 1
Reviewer 1 Report
A simplified murine model…
By Bao et al.,
The authors compared peritendinous adhesion formation in the novel and very simple longitudinal incision model with the tendon repair model which includes tendon transsection and subsequent suturing. They found that the tested parameters did not differ. Was for the transsection model also the plantar part of the tendon used? The conventional model should be added to the appendix figure. The novel model does not use an intrasynovial tendon, please comment on the occurence in intrasynovial adhesions. In humans, the tendon used for LCM here, corresponds rather with the plantar aponeurosis. The introduction is very short and not concise.
Abstract
abbreviation „LCM“ for longitudinal incision model, why not „LIM“?
line 35: optimize sentence and provide references for „few models“
Line 37: „Medicine and cell therapy“ I think the term „Medicine“ is not very informative here.
Line 42: Size, write plural: sizes
Line 56 and 84: delete surplus point
Line 57: „two experimental groups“, I think 3 groups are compared, see the sham group mentioned: sham surgery, line 64, please clarify!
Bring the skizz of the model in the main body oft he manuscript not only in the appendix, mark in A the tendon shown enlarged in B.
Why were male mice selected?
Vincula tendinum removed during modeling: why – movement is important for regular healing.
„The histological scoring system of adhesions was scored into grades 0-10 82 (Figure A2) and the histological scoring system of healing is scored into grades 0-3 in the“ improve the style!
Line 99: where is „AT“ introduced as an abbreviation?
2.5.
Why are different numbers of animals included in the respective groups (dop outs? ouliers?) please explain!
2.6. cursive like the other headings
3.1.
„Mice were…harvested“ write „tendon samples were harvested“
Line 136: explain early what the control group means (no tendon transsection? Only sham healing of surrounding connective tissue?)
Line 139: point doubled, line 129, 154, 177: points before figure citation.
Line 142: „explosion“ might not be the correct term
Line 148: „fiber parallel score“ how was it assessed with which categories?
Fig. 2: the fiber parallel score differs significantly – should be discussed in more detail
Fig. 3C-D: what is exactly shown in these subfigures?, add this information to the legend
Discussion
I think the first sentence should mean: PA is a complication of tendon rupture?
Line 206 and 207: write Achilles tendon
Line 210: write „ruptured“
Paragraph starting in line 224: it should be transposed rather tot he beginning since it is important fort he total understanding
Line 230: „Tayler SH etc.“ means Tayler et al.,
line 249: write plural „differences“
line 290: correct „fianl“
table A1: for collagen type III „®“ is lacking, how about the reference gene?? 2.3 mention the reference gene used!!
Reviewer 2 Report
In this research Bao et al. propose a simplified surgical method which imitates the clinical tendon adhesion without tendon transection and suture as an alternative to more classical methods (TRM).
The research is interesting and could represent a progress in animal experimentation modeling, however there are some issues that must be considered before publication:
1) There are several English grammar mistakes and typos throughout the text (e.g. lines: 20, 35, 65-66, 108, 129, 132, 142, 152, 180, 210, 230, 288, 290…) and some inaccuracies.
2) The authors should write in Materials and Methods how many animals in total they have used for each group since this information is not clearly deducible.
3) In “2.3. Gene Expression using Real-Time PCR” Table A1 is not cited within the text, meanwhile there is cited Figure A4 which is missing (I suppose it should be Table A1). Furthermore also the GADPH primer sequence should be mentioned in the table.
4) The results in “Gene Expression in Longitudinal Incision Model” should be exploited better. For example there is a significative difference between the surgically treated groups, where there is an upregulation of both genes, and the control. This result should be commented in the discussion. The authors should also better clarify this sentence “Thus, the gene expression of collagen I and collagen III represents the potential of adhesion formation” and discuss it in the Discussion section.
5) Figure A3: in panel (B) figures have been inverted, please correct.
Reviewer 3 Report
This study proposed a longitudinal incision model in the murine flexor tendon to imitate flexor tendon adhesion formation. The histological analysis, mechanical analysis and real-time RT-PCR were performed. The authors concluded that the longitudinal incision model may have a potential value with easy operation and good quality in future biomimetic investigation which imitates the clinical tendon adhesion without flexor tendon transection and subsequent suture.
General comments:
The study is interested and well organized. However, the detailed experimental setups and outcomes are expected to be provided.
Specific comments:
1) Lines 57-58, “8-10 week old male C57BL/6 mice were randomized into two experimental groups:”.
Is there any criterion to divide all mice into two groups? What is the sample size for each group? Please mention these important information after this sentence in the manuscript.
2) Lines 82-83, “The histological scoring system of adhesions was scored into grades 0-10 (Figure A2) and the histological scoring system of healing is scored into grades 0-3 in the field of cellularity, fiber parallel and fiber continuity. (Figure A3) [31,32].”
What are the physical meanings for the adhesion score, score of cellularity, score of fiber parallel, and score of fiber continuity? It was suggested to give more information to the interested readers who has no prior knowledge.
3) The range of motion tests were conducted to measure the metatarsophalangeal (MTP) joint flexion and gliding coefficient. It would be better if a figure can be prepared to demonstrate the range of motion tests. Additionally, the required experimental setups are needed to be mentioned.
4) What is Real-Time PCR? Please provide the detailed description and definition about Real-time PCR.
5) The biomechanical analysis was conducted by using the pullout tests. It would be better if a figure can be prepared to demonstrate the pullout tests.
6) How to measure the range of motion? Is there any measurement equipment or tool to be used? Is it possible to provide the detailed experimental curve for the range of motion?
7) In figure 5A, the stiffness is not accurately fit to the slope of the linear region of the load-displacement curve. Is there any criterion to be used for determining the slop of the stiffness?
Round 2
Reviewer 1 Report
The authors have addressed my previous comments: Accordingly the manuscript has been improved and the pecularity of the model is better understandable.
Idioms such as line 67: "What’s more" should be spelled out during proof correction. Please check the complete manuscript. Line 173: "Figure 2 D-G" please omit the blank following the style later used.
Reviewer 3 Report
The authors adequately responded to all comments and revised the manuscript. Consequently, I recommend that the manuscript be accepted for publication.